# Unglycosylated Soluble SARS-CoV-2 Receptor Binding Domain (RBD) Produced in *E. coli* Combined with the Army Liposomal Formulation Containing QS21 (ALFQ) Elicits Neutralizing Antibodies against Mismatched Variants

**DOI:** 10.3390/vaccines11010042

**Published:** 2022-12-25

**Authors:** Arasu Balasubramaniyam, Emma Ryan, Dallas Brown, Therwa Hamza, William Harrison, Michael Gan, Rajeshwer S. Sankhala, Wei-Hung Chen, Elizabeth J. Martinez, Jaime L. Jensen, Vincent Dussupt, Letzibeth Mendez-Rivera, Sandra Mayer, Jocelyn King, Nelson L. Michael, Jason Regules, Shelly Krebs, Mangala Rao, Gary R. Matyas, M. Gordon Joyce, Adrian H. Batchelor, Gregory D. Gromowski, Sheetij Dutta

**Affiliations:** 1Biologics Research and Development Branch, Structural Vaccinology Laboratory, Walter Reed Army Institute of Research, Silver Spring, MD 20910, USA; 2Emerging Infectious Diseases Branch, Walter Reed Army Institute of Research, Silver Spring, MD 20910, USA; 3Henry M. Jackson Foundation for the Advancement of Military Medicine, Inc., Bethesda, MD 20817, USA; 4U.S. Military HIV Research Program, B-cell Biology Laboratory, Walter Reed Army Institute of Research, Silver Spring, MD 20910, USA; 5Viral Diseases Branch, Walter Reed Army Institute of Research, Silver Spring, MD 20910, USA; 6Center for Infectious Disease Research, Walter Reed Army Institute of Research, Silver Spring, MD 20910, USA; 7U.S. Military HIV Research Program, Laboratory of Adjuvant and Antigen Research, Walter Reed Army Institute of Research, Silver Spring, MD 20910, USA

**Keywords:** vaccine, recombinant, *E. coli*, SARS-CoV-2, RBD

## Abstract

The emergence of novel potentially pandemic pathogens necessitates the rapid manufacture and deployment of effective, stable, and locally manufacturable vaccines on a global scale. In this study, the ability of the *Escherichia coli* expression system to produce the receptor binding domain (RBD) of the SARS-CoV-2 spike protein was evaluated. The RBD of the original Wuhan-Hu1 variant and of the Alpha and Beta variants of concern (VoC) were expressed in *E. coli*, and their biochemical and immunological profiles were compared to RBD produced in mammalian cells. The *E. coli*-produced RBD variants recapitulated the structural character of mammalian-expressed RBD and bound to human angiotensin converting enzyme (ACE2) receptor and a panel of neutralizing SARS-CoV-2 monoclonal antibodies. A pilot vaccination in mice with bacterial RBDs formulated with a novel liposomal adjuvant, Army Liposomal Formulation containing QS21 (ALFQ), induced polyclonal antibodies that inhibited RBD association to ACE2 in vitro and potently neutralized homologous and heterologous SARS-CoV-2 pseudoviruses. Although all vaccines induced neutralization of the non-vaccine Delta variant, only the Beta RBD vaccine produced in *E. coli* and mammalian cells effectively neutralized the Omicron BA.1 pseudovirus. These outcomes warrant further exploration of *E. coli* as an expression platform for non-glycosylated, soluble immunogens for future rapid response to emerging pandemic pathogens.

## 1. Introduction

The zoonotic transmission of betacoronaviruses such as severe acute respiratory syndrome coronavirus 1 (SARS-CoV-1) [1], Middle East respiratory syndrome [2], and SARS-CoV-2 have caused significant global morbidity and mortality [3]. Three years after the first documentation of SARS-CoV-2 transmission in humans, more than 600 million cases and 6.4 million mortalities have been attributed to COVID-19 [4]. Typical of its subfamily, SARS-CoV-2 utilizes a homotrimeric spike (S) protein to interact with the human angiotensin converting enzyme 2 (ACE2) receptor present on epithelial and endothelial cell surfaces for cellular entry [5,6]. Potently neutralizing antibodies against SARS-CoV-2 have been found to possess paratopes targeting the S protein, a realization that has guided the development of monoclonal antibody (mAb) therapies used in the clinic for pre-exposure or early prophylactic treatment [7,8,9]. Similarly, vaccines against SARS-CoV-2, including nucleoside-modified mRNA (developed by Moderna and Pfizer-BioNTech), replication-incompetent adenovirus (Johnson & Johnson and AstraZeneca), and recombinant S protein (Novavax and Sanofi), target the interaction of S protein with ACE2 [10]. 

The protomer of the S antigen is a 1273 amino acid (a.a.) protein organized into two major subunits: S1 (a.a. 14–685) and S2 (a.a. 686–1273) (Figure 1A) [6,11]. The S1 region contains an N-terminal domain (NTD; a.a. 14–305) and the receptor-binding domain (RBD; a.a. 319–541). RBD directly binds to human ACE2 [5,11,12], resulting in S1′s cleavage from S2 and the induction of a post-fusion conformation that mediates cell membrane fusion and entry into the host cell [5,13,14]. Host immune responses drive the selection of variants with mutations in the RBD subdomain that confer escape from neutralization, affording novel variants with the opportunity for rapid proliferation resulting in periodic surges in case numbers, particularly for immunocompromised patients [15]. The original Wuhan-Hu1 virus acquired the N501Y mutation that gave rise to the B.1.1.7 (Alpha) variant and its capacity to escape anti-Wuhan-Hu1 neutralizing monoclonal antibodies [7,16,17]. Variants B.1.351 (Beta; N501Y, K417N, and E484K) and B.1.617.2 (Delta; K417N, L452R, and T478K) accrued additional mutations and manifested an even greater resistance to neutralizing antibodies [16,18]. The B.1.1.529 (Omicron) variant, along with its subvariants BA.1–5, contain a large number of escape mutations in the RBD (G339D, S371L, S373P, S375F, K417N, N440K, G446S, S477N, T478K, E484A, Q493R, G496S, Q498R, N501Y, Y505H) [19,20] that enabled it to outcompete previous variants and become the most prevalent circulating variant in the United States in early 2022 [21,22,23]. Several cross-reactive mAbs targeting the RBD have been isolated [9,24,25], and efforts are underway to develop broadly protective vaccines [26,27]. 

Viral vaccine antigens typically have complex tertiary structures and require extensive post-translational modifications. SARS-CoV-2′s RBD contains 9 cysteine residues that form four disulfide bonds, as well as two N-linked (N331 and N343) and several putative O-linked glycosylation sites (T323, S325, T333, and T345) [28,29,30]. Eukaryotic platforms are therefore preferred over prokaryotic systems for the expression of viral antigens such as RBD, whose complex architectures require extensive disulfide networks and post-translational modifications [31]. Scale-up protein manufacturing in mammalian cells, however, requires licensed vectors, transfection reagents, patented stable cell-line development technologies, facilities, and experienced staff that may not be widely established in low- and middle-income countries. Prokaryotic expression systems like *E. coli*, which are well characterized and highly accessible, could enable the development of cost-effective and locally produced vaccine antigens needed for rapid, global pandemic responses. 

Our group has previously shown that malarial antigens with complex tertiary structures can be manufactured in *E. coli* under cGMP conditions [32,33]. When formulated with powerful adjuvants like AS01 (GlaxoSmithKline, or GSK) [34] or the Army Liposomal Formulation containing QS21 (ALFQ) adjuvant [33,35], *E. coli*-produced soluble proteins can elicit protective levels of antibodies. Throughout the early stages of the COVID-19 pandemic, we carried out proof-of-concept studies to determine if biologically active SARS-CoV-2 vaccine candidates could be produced using the *E. coli* expression system. Here, we report the expression of soluble SARS-CoV-2 RBD in *E. coli* and its induction of cross-neutralizing antibodies in a murine model when adjuvanted with ALFQ. The bacterial antigens were structurally, antigenically, and immunologically comparable to RBD expressed in mammalian cell lines. These results suggest that *E. coli*-produced soluble antigens could be feasible for the rapid and inexpensive manufacture of viral vaccines.

## 2. Materials and Methods

### 2.1. E. coli Expression of RBD

The nucleotide sequences corresponding to a.a. 319–542 of S protein from the original Wuhan-Hu1, Alpha (B.1.1.7), and Beta (B.1.351) variants of SARS-CoV-2 with a N-terminal 6x His tag were codon optimized for high level bacterial expression, commercially synthesized, and cloned into the pD451-SR vector (Atum, Newark, CA, USA). Genes were expressed in BL21(DE)_3_ *E. coli* cells (Novagen, Madison, WI, USA). Expression was initiated by a 100-mL starter lysogeny broth culture that was grown overnight at 37 °C. The culture was expanded the following day to a 1 L-scale using terrific broth; both cultures contained 50 µg/mL kanamycin. The culture was grown at 37 °C to an OD_600_ = 0.8 and induced with 1 mM isopropyl β-D-1-thiogalactopyranoside (IPTG) for 3 h. Biomass was recovered by centrifugation at 4000× *g* for 20 min at 4 °C. This paste was frozen at −80 °C until purification. 

### 2.2. E. coli RBD Purification and Refolding 

Cell paste in ~5 g batches was resuspended in resuspension buffer (RB) containing 20 mM Tris, 150 mM NaCl, 5 mM β-mercaptoethanol (pH 7.0). Cells were lysed by microfluidization using the M-110Y Microfluidizer (Microfluidics, Newton, MA) and centrifuged at 15,000× *g* for 20 min at 4 °C. The inclusion body pellet was washed with RB containing 1.5% Triton X-100 for endotoxin reduction, centrifuged at 15,000× *g* for 20 min at 4 °C, and solubilized in 25 to 50 mL equilibration buffer (EqB), which was RB that contained 7 M urea (pH 7.0). The protein solution was centrifuged again as above and the resultant supernatant was applied to a 5-mL Ni-NTA Superflow column (Qiagen, Germantown, MD, USA) mounted on an AKTA Purifier 100 (Cytiva, Marlborough, MA, USA). The column was pre-equilibrated with 5 column volumes (CV) of distilled water and 10 CV of EqB. After protein loading, the Ni-NTA column was washed with 10 CV of wash buffer containing 20 mM Tris, 300 mM NaCl, 40 mM imidazole, 5 mM β-mercaptoethanol, 7 M urea, 1% CHAPS (pH 7.0), then washed again with 10 CV of EqB. The protein was eluted using 5 CV of elution buffer (EB) containing 20 mM Tris, 150 mM NaCl, 250 mM imidazole, 5 mM β-mercaptoethanol, 7 M urea (pH 7.0). The eluate was diluted 4-fold in EqB to ~60 mM imidazole, then passed over a 5-mL Q Sepharose^®^ Fast Flow column (Cytiva) to reduce endotoxin. The flow-through was collected and dialyzed overnight at 4 °C in 20 mM Tris, 150 mM NaCl, 5% glycerol (pH 7.0), in a Slide-A-Lyzer™ G2 Dialysis Cassette, 10K MWCO (Thermo Fisher Scientific, Waltham, MA, USA). The purity of the final product was assessed by SDS-PAGE with Coomassie Blue staining and densitometric analysis using ImageJ [36]. The buffer-exchanged protein was centrifuged at 10,000× *g* for 15 min and concentrated to 1 mg/mL before aliquoting and storage at −80 °C. 

### 2.3. Mammalian Expression and Purification 

The Beta variant of RBD with a N-terminal 6x His tag was codon optimized and cloned into the proprietary pD2610-v5 vector (Atum) for high-level mammalian expression. For expression, transient transfection of the RBD-encoding plasmid into Expi293F™ cells was achieved using the ExpiFectamine™ 293 Transfection Kit per the manufacturer’s protocols (Thermo Fisher Scientific). Briefly, plasmid DNA purified using the EndoFree Plasmid Maxi Kit (Qiagen) was diluted in Opti-MEM I medium (1 µg/mL of final culture volume), sterile filtered, and combined with ExpiFectamine™ 293 reagent diluted in Opti-MEM I medium; this complex was incubated for 10 to 20 min at room temperature, then added to the cells diluted to a density of 3 × 10^6^ cells/mL. The culture was incubated in a 37 °C incubator with 85% humidity, 8% CO_2_ on an orbital shaker at 125 rpm for 18 to 22 h. Enhancer mixture was added according to manufacturer’s recommended ratio, and the culture was incubated for another 5 to 6 days. The culture supernatant was harvested and purified over a 2-mL Ni-NTA Superflow bed (Qiagen) in an Econo-Pac^®^ chromatography column (Bio-Rad, Hercules, CA, USA) equilibrated twice with 5 CV of 20 mM Tris, 150 mM NaCl, 2.5 mM β-mercaptoethanol (pH 7.0), washed with 5 CV of 20 mM Tris, 300 mM NaCl, 20 mM imidazole, 2.5 mM β-mercaptoethanol (pH 7.0), and eluted using 20 mM Tris, 150 mM NaCl, 250 mM imidazole, 2.5 mM β-mercaptoethanol (pH 7.0). The protein was finally dialyzed into phosphate-buffered saline (pH 7.4).

### 2.4. Endotoxin Content Quantitation

The endotoxin content of RBDs purified from *E. coli* was determined using a kinetic *Limulus* amebocyte lysate assay (Associates of Cape Cod, East Falmouth, MA, USA). Samples were diluted 1:10 in endotoxin-free water and serially diluted 11-fold down a certified endotoxin-free 96-well microplate (Associates of Cape Cod). Endotoxin standards ranging from 2 EU/mL to 0.125 EU/mL were prepared in endotoxin-free water using the Control Standard Endotoxin reagent (Associates of Cape Cod). Pyrochrome reagent was added to the sample in a 1:1 ratio and plates were read on a Synergy 4 microplate reader (BioTek, Winooski, VT, USA) at 30 °C for 1 h. The OD_405_ value was recorded every 2 min and subtracted from a blank well containing 0 EU/mL endotoxin. Endotoxin content (EU/mL and EU/μg of protein) was calculated per the manufacturer’s instructions. 

### 2.5. CR3022 Dot Blot

10 µg of recombinant RBD was applied on a nitrocellulose membrane, allowed to dry for 30 min, then blocked with 1% (*w*/*v*) casein in PBS (Thermo Fisher Scientific) for 30 min. The membrane was incubated with 10 µg/mL of mAb CR3022 [37,38] for 1 h, washed three times with 1x PBS containing 0.05% (*v*/*v*) polysorbate 20 (PBST), and incubated for 1 h with alkaline phosphatase-conjugated goat anti-human IgG (H + L) (SouthernBiotech, Birmingham, AL) diluted 1:2000 in the blocking buffer. The membrane was washed and finally developed using BCIP/NBT substrate (Sigma-Aldrich, St. Louis, MO, USA). 

### 2.6. Analytical Size Exclusion Chromatography

Analytical size exclusion chromatography was performed using the Acclaim™ SEC-1000 Size Exclusion Chromatography HPLC Column (Thermo Fisher Scientific) mounted on the Waters™ e2695 Separations Module (Waters, Milford, MA, USA) per the manufacturer’s operating guidelines. The running buffer consisted of 20 mM Tris, 150 mM NaCl, 5% glycerol (pH 7.0), and the flow rate was maintained at 0.5 mL/min. The elution time was calibrated using a protein molecular size standard. Approximately 50 μg of RBD was injected per run.

### 2.7. Mouse Immunizations

BALB/c mice (*n* = 4 per group) were intramuscularly immunized three times at 2-week intervals with RBD antigen formulated in 50 µL of ALFQ containing 20 µg 3D=PHAD^®^ and 10 µg QS-21 [39]. Three groups of mice received 10 µg of one of the three RBD variants (*E. coli*), a fourth group was given a mixture of all three proteins (~3.3 µg each), and a fifth group 10 µg of glycosylated Beta variant RBD produced in mammalian cells. All formulations were rotated on a rocker for 1 h and administered intramuscularly into alternating thigh muscles. Blood was collected by nicking the tail vein two weeks after the third vaccination (2WP3), at week 6 of the study, and six weeks after the third dose (6WP3) at week 10. 

### 2.8. Human ACE2-Fc and Monoclonal Antibody CR3022 Binding Assay

Biosensors used for biolayer interferometry (BLI) were hydrated in 1x PBS prior to the assay run. All assay steps were performed at 30 °C with agitation set at 1000 rpm in the Octet RED96 instrument (FortéBio, Fremont, CA, USA). Pre-hydrated AHC biosensors (FortéBio) were loaded with 30 µg/mL of ACE2-Fc for 120 s. Varying dilutions of RBD were associated for 180 s and dissociated for 300 s in 1x PBS to obtain binding kinetics. For mAb CR3022 binding, pre-hydrated AHC biosensors were loaded with 30 µg/mL of CR3022 for 120 s and varying dilutions of RBD were associated for 180 s and dissociated for 300 s in 1x PBS to obtain binding kinetics.

### 2.9. Novel Anti-RBD Monoclonal Antibody Binding Assay

HIS1K biosensors (FortéBio) were pre-hydrated in assay buffer (AB) containing 1x PBS, 0.002% (*v*/*v*) polysorbate 20, 0.01% (*w*/*v*) bovine serum albumin (pH 7.5). RBD at 5 µg/mL was loaded for 180 s. 5 µg/mL of mAbs WRAIR-2057, -2063, -2123, -2125, -2165 and -2173 [8] were associated for 120 s. The dissociation of these mAbs in AB was monitored for 300 s. 

### 2.10. ACE2 Binding Inhibition Assay

HIS1K sensors (FortéBio) pre-hydrated in AB were loaded with 5 µg/mL of individual RBD variants for 480 s. The 6WP3 sera pool at 1:50, 1:500, 1:5000, 1:10,000 and 1:20,000 dilutions were loaded for 120 s, then associated with 400 nM ACE2 for 120 s. Dissociation was monitored for 300 s. Percent inhibition of ACE2 binding by immune sera was calculated with respect to a naïve serum control pool (Appendix A). 

### 2.11. ELISA

Antigenicity ELISAs were carried out using 100 ng/well RBD variant coated onto a 96-well 4HBX Immulon (Thermo Fisher Scientific) plate at 4 °C overnight. Plates were washed thrice with 1x PBS containing 0.05% Tween-20, blocked for 1.5 h with 5% casein, and incubated with a 1:3 serial dilution of the monoclonal antibodies starting at 1 µg/mL for 2 h. The plates were washed, then incubated for 1 h with horseradish peroxidase-conjugated goat anti-human IgG diluted 1:4000 (Southern Biotech). Plates were washed and developed using 100 µL/well KPL substrate (SeraCare, Gaithersburg, MD, USA). After 1 h, 10 μL of 20% SDS stop solution was added and plates were read at 415 nm. The OD = 1 titers were determined from the readings on the Synergy 4 microplate reader (BioTek) using 4-parameter curve fitting. For immunogenicity ELISAs, 50 ng of each RBD variant was coated, and 1:3 serial dilutions of mouse sera starting at 1:300 dilution was incubated for 1 h. Goat anti-mouse IgG (H+L) horseradish peroxidase-conjugated (Southern Biotech) was used as the secondary antibody and the plates were developed as above. 

### 2.12. Pseudovirus Neutralization Assay

The neutralization assay for SARS-CoV-2 variants (Wuhan-Hu1, B.1.1.7, B.1.351, B.1.617.2 and B.1.1.529) was performed as previously described [40,41,42]. Infectivity and neutralization titers were determined using ACE2-expressing HEK293 target cells (Integral Molecular, Philadelphia, PA, USA). Test sera were diluted 1:40 in growth medium and serially diluted; 25 μL/well was then added to a white 96-well plate. An equal volume of diluted pseudovirus was added to each well, and plates were incubated for 1 h at 37 °C. Target cells were added to each well (40,000 cells/well), then plates were incubated for an additional 48 h. Relative light units were measured with the EnVision Multimode Plate Reader (Perkin-Elmer, Waltham, MA, USA) using the Bright-Glo Luciferase Assay System (Promega, Madison, WI). Neutralization dose-response curves were fitted by nonlinear regression using the LabKey Server. Final titers are reported as the reciprocal of the serum dilution necessary to achieve 50% inhibition of SARS-CoV-2 infectivity (ID_50_).

### 2.13. Statistical Analysis

A proof-of-concept vaccination study was conducted (*n* = 4). Both descriptive and summary statistics were applied. Differences between ELISA and neutralization titers of vaccine groups were compared using ANOVA followed by Tukey’s comparison test. *p* values < 0.05 were considered significant. Individual data and geometric means were plotted with GraphPad Prism 9.1 software.

## 3. Results

### 3.1. Wuhan-Hu1 and VoC RBD Expression in E. coli and Mammalian Cells

Wuhan-Hu1 RBD protein was expressed at a relatively high level following IPTG induction (Figure 1B, lanes 1 & 2). The majority of protein was localized in the inclusion body pellet (Figure 1B, lanes 3 & 4) and required 7 M urea for solubilization (Figure 1B, lane 7). The protein was purified to near homogeneity using Ni-NTA chromatography under reduced and denatured conditions (Figure 1B, lane 10) and refolded by one-step dialysis (Figure 1B, lane 11). The refolded Wuhan-Hu1 RBD was compared to mammalian cell-expressed Wuhan-Hu1 reference standard using a BLI-based binding assay (Appendix A). The association of bacterial Wuhan-Hu1 to hACE2-Fc and to a panel of SARS-CoV-2 mAbs (P2B-2F6, CC12.1, CC12.156, [43,44] and WRAIR-2125 [8]) and the SARS-CoV-1 mAb CR3022 [37] was comparable to the reference standard (Appendix A). The dissociation constants of *E. coli*-produced Wuhan-Hu1 RBD’s interaction with hACE2-Fc and mAb CR3022 were 3.5 nM and 6.1 nM, respectively, which are close to previously reported values for eukaryotic-expressed RBDs (4.7 nM for ACE2 and 6.2–6.3 nM for CR3022) [45,46] (Appendix A). These data suggest that *E. coli* may be used to express a biologically active form of Wuhan-Hu1 RBD protein. In late 2020, as the Alpha and Beta VoCs of SARS-CoV-2 emerged [21,22,23], we applied this expression and purification protocol to express these two additional RBD variants. Alpha and Beta SARS-CoV-2 RBDs showed similar purification (Figure 1C,D) and refolding profiles as Wuhan-Hu1. Product yields per gram wet cell paste were ~0.5 mg, ~0.2 mg, and ~0.15 mg RBD for Wuhan-Hu1, Alpha, and Beta variants, respectively. The endotoxin content of products ranged from 0.4 to 4 EU/μg. For immunological comparison we expressed a fully glycosylated version of the Beta RBD variant in Expi293F™ mammalian cells [30]. The mammalian product was purified by one-step Ni-NTA chromatography (Figure 2). 

### 3.2. Biochemical Characterization of RBD Variants Expressed in E. coli

Prior to immunological studies, a batch of *E. coli*-expressed RBD variants were compared to the mammalian-expressed RBD for biological and functional profiling. The 3 bacterial proteins showed ≥90% purity by densitometry, while the purity of the mammalian product exceeded 95%. A minor band corresponding to the ~50 kDa dimer of RBD was observed (Figure 1B–D and Figure 2A). Additionally, the mammalian control product migrated slower than the unglycosylated *E. coli* products due to its glycan decoration [30] (Figure 2A). According to BLI analysis, *E. coli* RBD variants had equivalent association and dissociation kinetics to the hACE2-Fc receptor as those of the mammalian-produced Beta RBD (Figure 2B). Dot blot of the *E. coli*-produced RBDs with a conformationally dependent SARS-CoV-1 mAb CR3022 provided additional confirmatory demonstration of proper folding (Figure 2C) [38]. A negative control protein, the full-length circumsporozoite protein of *Plasmodium falciparum*, did not react with mAb CR3022. Only trace reactivity to the mammalian-expressed RBD was observed. Analytical size exclusion chromatography indicated that the mammalian- and *E. coli*-expressed RBDs were homogeneous and eluted as a monomer peak around 12 min (Figure 2D). 

### 3.3. Antigenicity of the RBD Variants

Recombinant RBDs from *E. coli* were next tested for the presence of epitopes corresponding to potently neutralizing Wuhan-Hu1-specific SARS-CoV-2 human mAbs by BLI and ELISA [8] (Figure 3). Wuhan-Hu1 RBD produced in *E. coli* reacted with all 7 mAbs specific to different sites on the RBD (Figure 3A,C). The Alpha variant reacted broadly with the queried mAbs apart from WRAIR-2123, to which it displayed attenuated reactivity due to mutations in the target epitope [8]. *E. coli*- and mammalian-expressed Beta RBD reacted with mAbs WRAIR-2057, -2063, -2125, and -2151, but not mAbs WRAIR-2123, -2165 and -2173, which was also consistent with previous binding and neutralization analyses [8]. These observations suggest that *E. coli* and mammalian RBDs contain both variant-specific and cross-reactive mAb epitopes whose recognition is not impacted by protein glycosylation. The above-described trends in mAb reactivity observed by BLI were recapitulated by ELISA (Figure 3B). 

### 3.4. Immunogenicity of RBD Variants

Mice were vaccinated with three doses of RBD proteins formulated in a liposomal adjuvant containing QS21 (ALFQ). Sera were analyzed by ELISA and a pseudovirus neutralization assay (Figure 4A). Vaccine groups included monovalent formulations of *E. coli*-derived Wuhan-Hu1, Alpha, or Beta RBD, a trivalent mixture of the three *E. coli*-derived RBDs (3.3 µg each), and the monovalent mammalian-expressed Beta RBD control. Sera collected at week 6 (2 weeks post-third vaccination, or 2WP3) and at week 10 (6WP3) were analyzed by ELISA (Figure 4B,C). Pre-immune serum dilutions failed to achieve OD=1 even at the lowest dilution (not plotted). In contrast, the 2WP3 GM titers for the monovalent and trivalent RBD vaccines exceeded 10,000 (Figure 4B). *E. coli*-derived Wuhan-Hu1 had the lowest immunogenicity, while the trivalent vaccine showed similarly elevated titers across plate antigens, indicative of the induction of broadly reactive antibodies. *E. coli*-derived Beta variant titers were comparable to those of the mammalian-expressed Beta variant, suggesting that *E. coli* was able to produce highly immunogenic viral antigens. By week 10, the titers showed a median 2- to 3-fold drop in magnitude (Figure 4C). Sera from six weeks post-third immunization were analyzed for in vitro inhibition of RBD binding to ACE-2 receptor (Figure 4D). Relative to the pre-immune controls (0% inhibition; not plotted), immune mouse serum pools from RBD-vaccinated mice inhibited the binding of ACE2 to the homologous RBD. Although inhibition by Wuhan-Hu1 was slightly lower, as expected due to its lower titers, ~100% inhibition was achieved at 1:50 serum dilution for all other vaccines. 

### 3.5. Pseudovirus Neutralization

Sera from mice vaccinated with *E. coli*-derived Wuhan-Hu1, Alpha, Beta, trivalent and mammalian-expressed Beta RBD were tested for inhibition of homologous and heterologous pseudoviruses (Figure 5). Because 50% neutralization (ID_50_) titers for pre-immune sera were <1:100 (not plotted), a geometric mean neutralization titer above 1:100 was considered positive. At 2WP3, *E. coli*- and mammalian-expressed Beta RBDs were compared for immunological similarity. Although the *E. coli* product showed lower Alpha pseudovirus neutralization, there were no significant differences in ID_50_ of these two vaccines across all five pseudoviruses (Figure 5, top panels). Despite one mouse in the *E. coli* Wuhan-Hu1 group that failed to neutralize, the geometric mean ID_50_ of all vaccine formulations indicated positive neutralization of the vaccine-matched Wuhan-Hu1, Alpha and Beta pseudoviruses. Against the vaccine mismatched Delta peudovirus, all vaccines showed inhibitory responses (ID_50_ > 1000), with the *E. coli*-derived Wuhan-Hu1 exceeding neutralization by the Beta RBD expressed in mammalian cells. Against the Omicron BA.1 pseudovirus, only the *E. coli* and mammalian Beta RBD showed neutralization (ID_50_ > 1000). Surprisingly, the monovalent Beta RBDs also outperformed the trivalent formulation. Neutralization patterns at 6WP3 showed a similar trend as those of the 2WP3 timepoint, with a median 1.5 to 2-fold drop in individual ID_50_ titers (Figure 5, bottom panels). 

## 4. Discussion

Prokaryotic expression of SARS-CoV-1 RBD (a.a. 318–510 [47] and a.a. 450–650 [48]) first established the feasibility of using *E. coli* to produce biologically active RBD-based immunogens. Subsequent *E. coli* expression of SARS-CoV-2 Wuhan-Hu1 variant RBD (a.a. 319–640 [49], a.a. 318–510 [50], a.a. 319–534 [51], a.a. 371–541 [28], a.a. 319–541 [52], a.a. 330–583 [53], and a.a. 519–541 [31]) further evinced the utility of this platform for coronaviral immunogen production. Across studies, the solubility of RBD in the *E. coli* cytosol was observed to be very poor, which necessitated extraction using a chaotropic agent followed by in vitro refolding. The reducing environment of the *E. coli* cytoplasm and inability of *E. coli* to chaperone correct disulfide bond formation [31] likely accounts for the relatively low yields observed across studies. Despite this, refolded RBDs have demonstrated binding to SARS-CoV-2 antibodies [49] and the human ACE2 receptor [31,50], and their overall structure was similar to mammalian-derived products [31,53]. *E. coli* RBD vaccination in small animals with multiple adjuvants, including Al(OH)_3_ + CpG ODN [51], Sigma Adjuvant System (Sigma-Aldrich) [47], TiterMax Gold (Sigma-Aldrich) [50], Alum [28], and Saponin [54], have shown high immunogenicity and the induction of neutralizing antibodies. Soluble proteins that are generally considered weak immunogens require formulation with potent adjuvants [55]. The ALFQ adjuvant used was recently employed in a soluble protein vaccine against malaria [33]; our data provides additional confirmation of the utility of this adjuvant for soluble immunogens. Overall, an accumulating body of evidence suggests that *E. coli*-produced viral antigens can match the vaccine potential of equivalent mammalian cell products. Additionally, the absence of surface glycans on *E. coli*-produced vaccines may reduce the risk of developing autoreactive antibodies against self-carbohydrate moieties [28]. As novel variants with the capacity to escape extant vaccines continually emerge, *E. coli* could provide a highly cost-effective platform for response against SARS-CoV-2 and other viral pathogens like influenza [56,57] and dengue fever [58,59,60].

ID_50_ titers >100 have been generally associated with protective responses in humans [61,62]. Evaluation of approved human vaccines in mice have reported ID_50_ titers in the 1000–10,000 range (Moderna’s mRNA-1273 vaccine [63] and AstraZeneca’s adenoviral vectored vaccine AZD1222 [64]). While only head-to-head comparison of vaccines can truly be definitive, we report neutralization titers similar to those reported for these licensed SARS-CoV-2 vaccines. Cross-neutralization of heterologous variants is another key requirement for next-generation coronavirus vaccines. The capacity of mismatched SARS-CoV-2 variants to escape monovalent vaccines has led to the development of heterologous prime-boost vaccine regimens [65,66]. In the present study, monovalent Wuhan Hu-1, Alpha, and the trivalent immunogens all failed to neutralize the Omicron BA.1 pseudovirus. However, *E. coli*-expressed and mammalian-expressed Beta variant antibodies cross-neutralized Omicron BA.1. This observation aligns with prior reports that infection or vaccination with the Beta variant induces potent, cross-neutralizing antibodies [25,67]. Possibly, the Beta variant RBD expressed in *E. coli* favorably presents conserved epitopes to the immune system due to the absence of glycosylation. 

Trimeric S protein, virus-like particles [41], nanoparticles [68], and T-cell epitope-enriched bacterial carriers [52] have been shown to effective vaccine platforms against SARS-CoV-2. Our data suggests that soluble, prokaryotic-expressed proteins may also be effective in eliciting broad immunity. Along these lines, a soluble RBD protein-based SARS-CoV-2 vaccine is currently being evaluated for efficacy in India [69]. The purification and refolding methodology described here translated across three variants of concern; while the process needs to be further optimized, a similar method may be useful for the manufacture of future SARS-CoV-2 variant RBD vaccines. 

## 5. Conclusions

A bacterial expressed viral immunogen was found to be equivalent to a mammalian-expressed immunogen across several evaluative criteria. The expression and purification protocol described here allowed for the production of several SARS-CoV-2 variants. Our data warrants further exploration of *E. coli* expression system for producing low-cost viral vaccines in resource-poor countries. 

## Figures and Tables

**Figure 1 vaccines-11-00042-f001:**
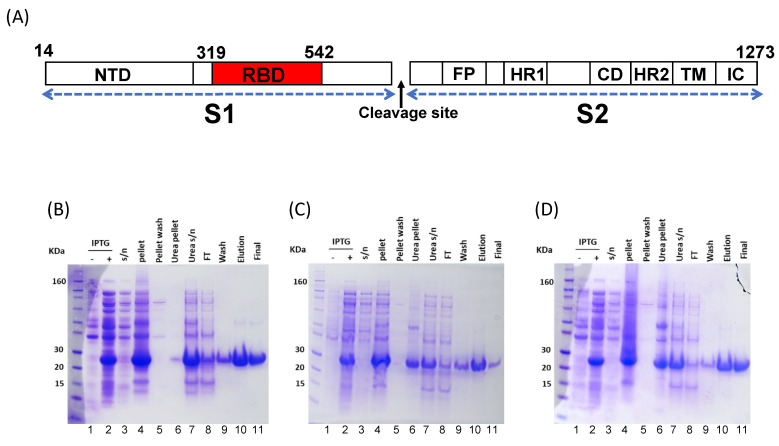
An overview of the domain organization of SARS-CoV-2′s RBD and its purification from the *E. coli* expression system. (**A**) Domain architecture of the spike (S) protein of SARS-CoV-2 (not to scale). The subdomain 1 (S1) contains the N-terminal domain (NTD, a.a. 14−305) and the receptor binding domain (RBD, a.a. 319−542). The receptor binding motif (a.a. 438−506) is contained within the RBD. Subdomain 2 (S2) contains the fusion peptide (FP) and two heptad repeats (HR1, a.a. 908 −985; HR2, a.a 1163−1214), which are separated by the connecting domain (CD). The S protein is anchored within the viral membrane by a transmembrane membrane helix (TM); the intracellular domain (IC) remains inside the virus. The S1 subunit is released post-cleavage during the fusion process. (**B**) Expression and purification of Wuhan-Hu1 variant, (**C**) Beta variant, and (**D**) Alpha variant RBDs from *E. coli*. Lanes 1 & 2, −/+ IPTG: whole-cell bacterial lysates; lane 3, s/n: supernatant resulting from microfluidization and centrifugation; lane 4, pellet: pellet resulting from cell lysis (inclusion body); lane 5, inclusion body wash; lane 6, pellet resulting from urea solubilisation of inclusion body; lane 7, urea solubilized inclusion body (input for Ni-NTA); lane 8, FT, flow-through from Ni-NTA; lane 9, wash Ni-NTA; lane 10, elution Ni-NTA; lane 11, final RBD product post-dialysis.

**Figure 2 vaccines-11-00042-f002:**
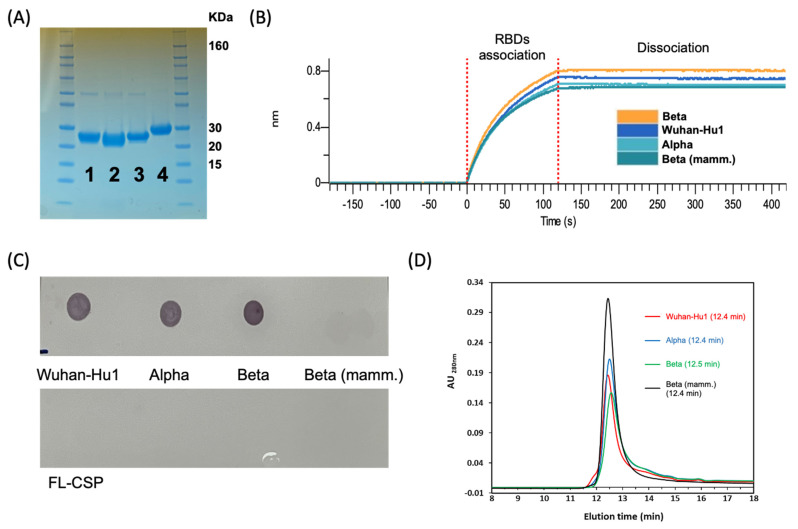
Biochemical characterization of recombinant RBD. (**A**) SDS-PAGE analysis of *E. coli*- and mammalian-derived recombinant RBDs (~26 kDa). Lane 1: Wuhan-Hu1 variant; lane 2: Alpha variant; lane 3: Beta variant; lane 4: mammalian-expressed Beta variant (~30 kDa). (**B**) RBD functionality determined by association to the human-ACE2 receptor using BLI analysis. (**C**) Dot blot of *E. coli*- and mammalian-produced RBDs using the SARS-CoV-1 anti-RBD antibody, CR3022. Full-length *P. falciparum* CSP (FL-CSP) was used as the negative control. (**D**) Analytical size exclusion chromatography indicates that all purified RBDs were homogenous.

**Figure 3 vaccines-11-00042-f003:**
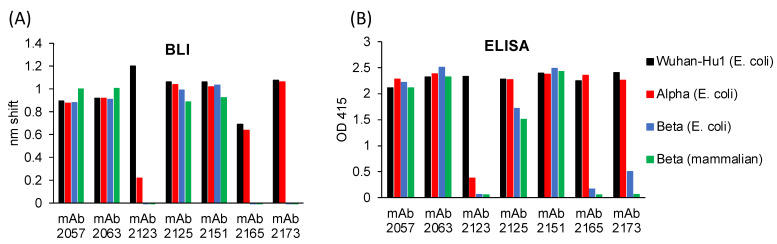
Antigenicity of recombinant RBDs. Novel human mAbs produced against Wuhan-Hu1 (mAbs WRAIR-2057, -2063, -2123, -2125, -2151, -2165 and -2173) were tested for reactivity to recombinant Wuhan-Hu1, Alpha, and Beta variant RBDs from *E. coli* and the Beta variant RBD from mammalian cells using (**A**) BLI and (**B**) ELISA. MAbs WRAIR-2057, -2063, -2125, and -2151 showed broad reactivity, whereas mAbs WRAIR-2123, -2165, and -2173 were variant-specific. (**C**) Epitopes for WRAIR-2151 (PDB: 7N4M), WRAIR-2123, WRAIR-2125 (PDB 7N4L), WRAIR-2165, WRAIR-2173 (PDB: 7N4J), WRAIR-2057 (PDB: 7N4I), and WRAIR-2063 (PDB: 8EOO) are colored red, yellow, blue, pink, orange, green, and purple, respectively.

**Figure 4 vaccines-11-00042-f004:**
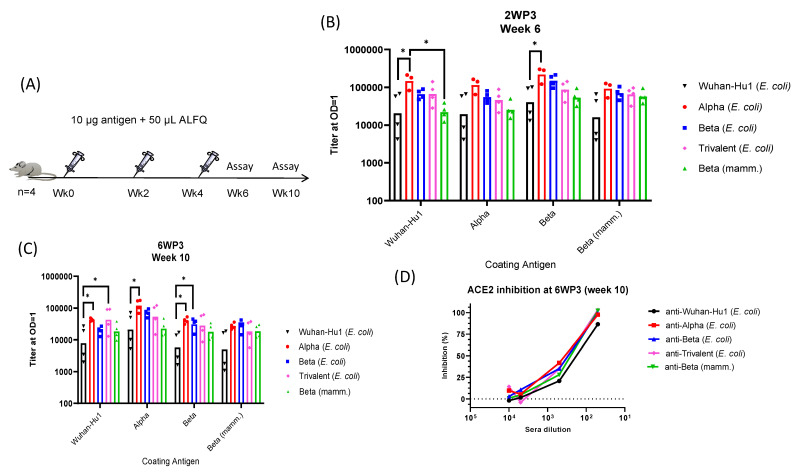
Immunogenicity of recombinant RBDs. (**A**) Schematic of the mouse study design. (**B**) ELISA (OD = 1 titer) against Wuhan-Hu1, Alpha, and Beta coat antigens at 2WP3 (week 6) and at (**C**) 6WP3 (week 10) are shown. Individual mouse titers are indicated for each vaccine group (*E. coli* Wuhan-Hu1, black inverted triangles; *E. coli* Alpha, red circles; *E. coli* Beta, blue squares; trivalent, pink diamonds; mammalian Beta, green triangles), while geometric mean titers are indicated by each bar. (**D**) Inhibition of homologous RBD binding to human ACE2 by serum pool collected at week 10 (6WP3). Inhibitions are relative to inhibition by normal mouse serum at a 1:50 dilution (adjusted to 0%; not plotted). Significant *p*-values between groups are shown with * (*p* < 0.05).

**Figure 5 vaccines-11-00042-f005:**
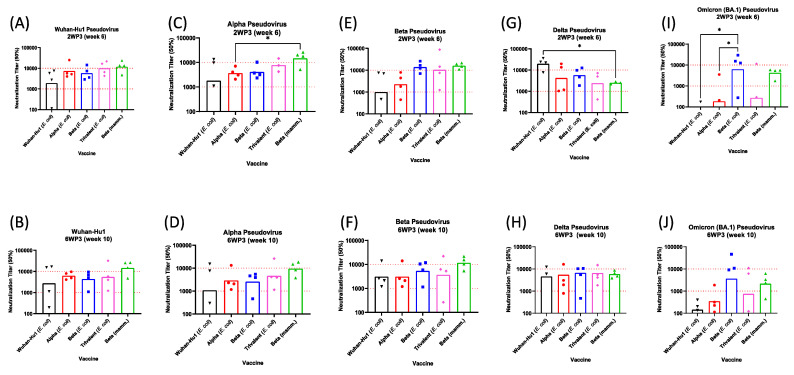
Pseudovirus neutralization titers induced by vaccination with recombinant RBDs. Individual mouse ID_50_ titers are indicated for each vaccine group (*E. coli* Wuhan-Hu1, black inverted triangles; *E. coli* Alpha, red circles; *E. coli* Beta, blue squares; trivalent, pink diamonds; mammalian Beta, green triangles), while geometric mean titers are indicated by each bar. Mouse sera at week 6 (2WP3, top panels) and week 10 (6WP3, bottom panels) were tested in pseudovirus neutralization assays against Wuhan-Hu1 (**A**,**B**), Alpha (**C**,**D**), Beta (**E**,**F**), Delta (**G**,**H**), and Omicron BA.1 (**I**,**J**) pseudoviruses. Pre-immune sera showed ID_50_ titers <40 across all pseudoviruses (not plotted). Significant *p*-values between groups are shown with * (*p* < 0.05).

## Data Availability

Authors will provide the raw data to any publicly available datasets as request.

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
