# Peer review of "Unglycosylated Soluble SARS-CoV-2 Receptor Binding Domain (RBD) Produced in E. coli Combined with the Army Liposomal Formulation Containing QS21 (ALFQ) Elicits Neutralizing Antibodies against Mismatched Variants"

_vaccines, 2022, doi:10.3390/vaccines11010042_

Round 1

Reviewer 1 Report

The authors reported that E.coli-expressed RBD of SARS-CoV-2 spike protein can reserve similar structure of mammalian-expressed RBD and can induce neutralizing antibody titers against SARS-CoV-2 pseudovirus. It is not easy to keep the right conformation of protein after refolding. The technique is potentially applied in the future outbreak of emerging infectious diseases. However, the manuscript needs to be improved before accepted for publication.

Commends:

1.    Lack of clear title in figure 1.

2.    The protein purity, yield and endotoxin levels need to be provided.

3.    Lack of purification figure of mammalian cell expressed RBD.

4.    Fig. 2C, please provide dot blot of mammalian RBD.

5.    The statistical analysis needs to be performed in the figure 5 and 6. Both figures can be integrated into one figure.

6.    The neutralization titer between Beta RBD (E. coli) and Beta RBD (mammalian) in the alpha variant may be different. Please discuss the observation.

7.     Please check labels in all figures to have the same font size and types.

8.    The English written should be improved.

Author Response

Reviewer-1

Lack of clear title in figure 1.

Response: We have revised to make the title for Figure 1 clearer.

The protein purity, yield and endotoxin levels need to be provided.

Response: Purity of bacterial RBD products was estimated by densitometry and was now added to the results (line 287). Also include now is the protein yield per gram paste (line 246) and the residual endotoxin levels for the three bacterial products (line 247).

Lack of purification figure of mammalian cell expressed RBD.

Response: This manuscript was aimed at highlighting expression of RBD in the bacterial system. Since the mammalian product was purified from a culture supernantant only the final SDS-PAGE profile of the product was included (Lane 4 in Figure 2A). The final purity of the mammalian product was close to 100% by densitometry and this data has now been added (line 287).

Fig. 2C, please provide dot blot of mammalian RBD.

Response: As advised, the dot blot for E. coli and mammalian RBD was repeated and the mammalian RBD was included. The newly added figure 2C showed poor reactivity of the mammalian RBD to mAb CR3022 but the three bacterial products showed excellent reactivity (line 292). It is possible that the glycosylated mammalian product did not bind well to nitrocellulose, regardless the requested blot has been added.

The statistical analysis needs to be performed in the figure 5 and 6. Both figures can be integrated into one figure.

Response: We have revised both Figs 5 and 6 and combined these into one figure. Due to the low number of mice per group (n=4) only a few differences were statistically significant. These statistical analyses and significant P values are now plotted in the revised Fig 5.

The neutralization titer between Beta RBD (E. coli) and Beta RBD (mammalian) in the alpha variant may be different. Please discuss the observation.

Response: As pointed to by the reviewer, the neutralization of the Alpha variant by the E. coli vs. mammalian Beta RBD sera has been noted in the results (line 339).

Please check labels in all figures to have the same font size and types.

Response: We have harmonized figure fonts.

The English written should be improved.

Response: We have gone through the manuscript and corrected for grammar and several changes were made to improve readability.

Reviewer 2 Report

The manuscript by Dutta and coworkers reports a new vaccine formulation consisting of an unglycosylated RBD antigen and a liposomal ALFQ adjuvant. The RBDs of the original Wuhan-Hu1 and the alpha and beta variants were prepared in E. coli expression system. Vaccination of mice with the vaccine candidate induced RBD-specific polyclonal antibodies that prevented ACE2 binding to RBD in vitro, and neutralized homologous and heterologous pseudoviruses. The authors also observed that all the vaccines induced cross-neutralizing antibodies to the non-vaccine Delta variant, however only the beta RBD vaccine induced cross-neutralizing antibodies to the Omicron BA.1 variant. In comparison, the E. coli-produced unglycosylated recombinant RBDs seemed to be similar to the glycosylated RBDs produced in mammalian cell lines as vaccine antigens, i.e., RBD glycosylation did not seem to affect epitope recognition, suggesting the feasibility of using the E. coli system to produce RBD antigens.

However, it is unclear how the ALFQ adjuvant affected the RBD-induced immune responses, including IgG titers, durability, and homologous and heterologous neutralization. It will be helpful to explain why it is needed (instead of other adjuvants) and whether it is at its optimal dose. Proper control groups should be included.

Author Response

Reviewer 2:

It is unclear how the ALFQ adjuvant affected the RBD-induced immune responses, including IgG titers, durability, and homologous and heterologous neutralization. It will be helpful to explain why it is needed (instead of other adjuvants) and whether it is at its optimal dose.

Response: We have added additional information supporting the use of ALFQ and ALFQ-like adjuvants for soluble antigens that have low immunogenicity (line 369). Since we did not compare ALFQ to ther adjuvants in this study we do not want to speculate how our RBD antigens formulated in alternative adjuvants may have performed. The dose of ALFQ (50 microliter) per vaccine has been standardized or for all our mouse studies. This represents 1/20th of a human dose (1 ml).

Proper control groups should be included.

Response: The adjuvant-only control group was not included in our mouse study. However, we have pre-immune serum as the control in all our functional assays. Percent inhibition of ACE2 binding (Fig 4D) was calculated with respect to the pre-immune serum and pseudovirus neutralization assays showed pre-immune serum ID50 titers <40 which were not plotted. Negative control wells were also included in the ELISA (Fig 4) and dot blot (Fig 2C).

Round 2

Reviewer 1 Report

The authors have answered all of my concerns.

Reviewer 2 Report

The revision looks fine. I support its publication in Vaccines